# Management of Common Infections in German Primary Care: A Cross-Sectional Survey of Knowledge and Confidence among General Practitioners and Outpatient Pediatricians

**DOI:** 10.3390/antibiotics10091131

**Published:** 2021-09-20

**Authors:** Peter Konstantin Kurotschka, Elena Tiedemann, Dominik Wolf, Nicola Thier, Johannes Forster, Johannes G. Liese, Ildiko Gagyor

**Affiliations:** 1Department of General Practice, University Hospital Würzburg, 97080 Würzburg, Germany; Tiedemann_e@ukw.de (E.T.); dominik.wolf95@hotmail.de (D.W.); nicolathier@icloud.com (N.T.); Gagyor_I@ukw.de (I.G.); 2Institute for Hygiene and Microbiology, University of Würzburg, 97080 Würzburg, Germany; Forster_J1@ukw.de; 3Department of Pediatrics, University Hospital Würzburg, 97080 Würzburg, Germany; liese_j@ukw.de

**Keywords:** infectious diseases management, general practitioner, pediatrician, primary care, outpatient, antibiotic use, antimicrobial resistance, antimicrobial stewardship, survey, knowledge

## Abstract

Outpatient antibiotic use is closely related to antimicrobial resistance and in Germany, almost 70% of antibiotic prescriptions in human health are issued by primary care physicians (PCPs). The aim of this study was to explore PCPs, namely General Practitioners’ (GPs) and outpatient pediatricians’ (PDs) knowledge of guideline recommendations on rational antimicrobial treatment, the determinants of confidence in treatment decisions and the perceived need for training in this topic in a large sample of PCPs from southern Germany. Out of 3753 reachable PCPs, 1311 completed the survey (overall response rate = 34.9%). Knowledge of guideline recommendations and perceived confidence in making treatment decisions were high in both GPs and PDs. The two highest rated influencing factors on prescribing decisions were reported to be guideline recommendations and own clinical experiences, hence patients’ demands and expectations were judged as not influencing treatment decisions. The majority of physicians declared to have attended at least one specific training course on antibiotic use, yet almost all the participating PCPs declared to need more training on this topic. More studies are needed to explore how consultation-related and context-specific factors could influence antibiotic prescriptions in general and pediatric primary care in Germany beyond knowledge. Moreover, efforts should be undertaken to explore the training needs of PCPs in Germany, as this would serve the development of evidence-based educational interventions targeted to the improvement of antibiotic prescribing decisions rather than being focused solely on knowledge of guidelines.

## 1. Introduction

In the last decades, concerns around the growing health and economic burden of antimicrobial resistance (AMR) urged the international public health community and the national health systems to action [1,2]. To face AMR, reducing unnecessary antibiotic use is established as the leading strategy, with the aim to preserve the efficacy of these drugs [1,2], both in the hospital and in the outpatient sector [3]. In addition, a reduction in antibiotic use may be associated with a lower risk of adverse outcomes, such as overtreatment of self-limiting infections, repeat consultations, side effects and rising costs [4]. According to the European Surveillance of Antimicrobial Consumption Network (ESAC-net), community antibiotic use in Europe in 2019 ranged between 32.4 Defined Daily Doses (DDDs)/1000 inhabitants/day (DIDs) in Greece and 8.9 DIDs in the Netherlands [5]. Compared to other European countries, outpatient antibiotic use in Germany is relatively low but the proportion of prescribed second-line antibiotics is high [5]. The majority of antibiotic drugs are prescribed in the outpatient setting. General Practitioners (GPs) and Internal Medicine Specialists who work as GPs are responsible for 59% of total antibiotic prescriptions; outpatient pediatricians (PD) for 9% [6,7]. In Germany, GPs and Internal Medicine Specialists who work as GPs are responsible mainly for providing primary care to adult and adolescent (12 years old or older) patients, whereas outpatient pediatricians care for patients until the age of 12 [8].

Prescriptions are a complex phenomenon and antibiotic prescribing is potentially influenced by clinical but also by multiple extra-clinical intrinsic (previous clinical practice, physician’s knowledge, confidence, fear, complacency to patients’ demands, responsibility of others) and extrinsic factors (guideline implementation, time pressure, workload, financial incentives, pharmaceutical companies, organizational level) [9]. Previous studies among GPs and outpatient pediatricians in Germany could show that knowledge and awareness about antibiotic use and resistance is high [10,11]. Nevertheless, a physician’s antibiotic prescribing behavior varies considerably within the country and between individual prescribers [6,12] and antibiotics are often prescribed unnecessarily and contrary to guideline recommendations [13,14]. The reason for this phenomenon is still not clearly understood. Understanding the factors that influence primary care physicians´ antibiotic prescribing practices in Germany and the lessons learned could inform antibiotic stewardship programs targeted to the needs of those physicians. Therefore, the aim of this study was to explore GPs’ and PDs’ knowledge of guideline recommendations on rational antimicrobial treatment in primary care and to explore the determinants of confidence in treatment decisions. Additionally, we analyzed the perceived need for training in this topic.

## 2. Results

As shown in Figure 1, out of 3753 physicians, 1311 completed the survey (overall response rate = 34.9%). Of 2103 letters sent to GPs, 11 letters were returned as undeliverable, reducing the effective number to 2092; 630 GPs responded to the survey (response rate = 30.1%). Likewise, 1680 PDs were contacted and 19 letters were undeliverable; 681 out of 1661 responded to the survey (response rate = 41%).

Only a few physicians participated in the online survey (GPs: n = 25/630, 4%; PDs: n = 34/681, 5%). 

Participants’ characteristics are shown in Table 1. The majority of participants were male (62.4% of GPs and 56.1% of PDs), with some not disclosing their gender (GPs: 15.6%; PDs: 11.0%). GPs were more likely to be older and to work in rural areas than PDs and most of them declared to work in group practices (defined as practices with two or more physicians).

Overall, knowledge scores were high in both GPs and PDs: on average, 17 out of 20 questions were answered correctly by PDs and 15 out of 19 questions by GPs. Five out of the seven knowledge items which were identical in both questionnaires were answered correctly by at least 85% of participants (Table 2). The logistic regression analysis showed that working as a PD was significantly associated with more correct answers to the questions on tonsillitis management (OR 0.28, 95%CI 0.18–0.43 and OR 0.46 95%CI 0.35–0.60, *p* < 0.001) and otitis media (OR 0.30, 95%CI 0.19–0.47, *p* < 0.001) compared to working as a GP. Slightly more correct answers were also given by PDs to a question focusing on pneumonia etiology (OR 0.68, 95%CI 0.48–0.95, *p* = 0.023). Conversely, GPs showed significantly higher knowledge scores on erythema migrans (OR 3.21, 95%CI 2.27–4.54, *p* < 0.001) and development of AMR (OR 1.34, 95%CI 1.09–1.71, *p* = 0.007) (Table 2).

Table 3 shows the level of confidence in antibiotic use and the factors that participants believe to influence their decisions to prescribe antibiotics. The perception of confidence was similar across all items on the four categories: indication, dose, duration and choice of antibacterial agent. Overall, GPs as well as PDs felt very confident in prescribing antibiotics, with PDs feeling significantly more confident: on a 7-point Likert scale, the mean score of confidence was 5.8 for GPs and 6.3 for PDs (Cohen´s d 0.5, 95% CI 0.4–0.6, *p* < 0.001). In prescribing antibiotics, the highest rated influencing factors in both groups were guideline recommendations (mean score = 5.6 for GPs and 6.3 for PDs, *p* < 0.001) and own clinical experience (mean score = 5.9 for GPs and 6.0 for PDs, *p* < 0.05). Patients’ wishes and demands, time pressure and the expectation that symptoms could worsen without antibiotics were declared to play a minor role in prescribing decisions.

No association was detected between the physician’s knowledge and the subjective feeling of confidence in antibiotic use among either GPs (Spearman’s correlation coefficient: r = −0.03, *p* = 0.48) or PDs (r = −0.02, *p* = 0.66).

As shown in Table 4, 77.2% of GPs and 84.8% of PDs reported that they participated in at least one specific training course for antibiotic therapy in the past 3 years. At the same time, 81.4% of GPs and 66.8% of PDs wanted more training on this topic. Consistently, as shown in Table 5, GPs had lower mean scores of perceived training sufficiency than PDs (3.7 vs. 4.6, respectively, *p* < 0.001).

## 3. Discussion

### 3.1. Comparison to Existing Literature

With this study, we performed a large-scaled survey among primary care physicians in general and pediatric practices in southern Germany that explored physicians’ knowledge and confidence, as well as its determinants of treatment decisions in patients with common infectious diseases.

Participating physicians achieved high scores in knowledge questions with slight differences between GPs and PDs. This finding is consistent with previous studies [10,11] but apparently contradicts the fact that in Germany, a relevant number of outpatient antibiotics are prescribed not in line with guideline recommendations [13]. For instance, a recent observational study conducted in Germany included more than 1.4 million patients from 1237 general and 239 pediatric practices between 2015 and 2019 and found that, despite being discouraged by a national guideline, 57% of adult and 23% of pediatric patients diagnosed with acute bronchitis received antibiotics. Knowledge questions in our survey explored guideline-knowledge which is relatively easy to retrieve and the response patterns could have been influenced by socially desirable answers [15]. In line with this interpretation is that GPs and PDs in our study declared to be influenced in their antibiotic prescribing decisions mainly by guideline recommendations and clinical experience (*p* <0.001 and <0.05, respectively). Patients’ wishes, expectations and demands, as well as the expectation that symptoms could worsen without antibiotics, were declared not to be relevant in their management decision-making process (*p* < 0.0001). These findings are in contrast with previous studies that show how antibiotic prescribing decisions are rather influenced by multiple psychologically- and socially-rooted factors than being the result of pure scientific reasoning [16]. Physicians’ prescribing behavior of antibiotics is influenced by several factors, such as patients’ expectations [17,18] and safety concerns [19]. Patients tend to be more dissatisfied with consultation in low prescribing practices [20], although, a recent qualitative study found that patients’ beliefs are evolving, probably due to rising awareness and antibiotics are perceived as medicines that should be prescribed when appropriate [21]. Physicians’ perception of patients’ expectations and hopes for antibiotics were found to be strongly associated with antibiotic prescribing [22,23], even if the clinician judges that antibiotics are not indicated [24]. Our findings could indicate that in our context, there is a lack of awareness among clinicians of the influence of patients’ expectations on their own prescribing behavior. Therefore, as suggested elsewhere [4], future training for physicians, as well as future studies in Germany, should take into proper consideration the influence of the doctor-patient relationship on the physicians’ decision-making process.

We showed that PDs were more likely to correctly answer questions regarding upper respiratory tract infections than GPs. This finding may be explained by the fact that respiratory infections are far more common in pediatric primary care than in general practice [25,26] and therefore, PDs could be more interested in the topic. Consistently, the response rate to our survey was 40.1% among PDs (versus 30.4% among GPs).

We found that 77.2% of GPs and 84.8% of PDs stated to have attended at least one training course addressing antimicrobial treatments, differently from previous reports [27]. At the same time, the majority of GPs and PDs felt they would need more training. This was despite the high scores they achieved in knowledge questions and the high level of confidence in antibiotic use. The reason for these conflicting results could be hypothesized, again, as having roots in social desirability: attending training courses is desirable, as is a high level of confidence, as well as expressing the need for more training.

### 3.2. Strengths and Limitations

A low response rate is a common problem in survey research and research shows that this is true, especially when surveys are conducted among physicians rather than among non-physicians [28,29]. In our case, our questionnaire showed a good acceptability due to a low number of clearly stated items. Consequently, in our study, we could reach 1311 physicians (overall response rate = 34.9%) with a low proportion of missing values (only 2 out of 630 GPs (0.3%) and 3 out of 681 PDs (0.4%) completed less than half of the questionnaire). This after only one postal invitation and one postal reminder and without any incentives given to responding physicians—incentives that could have led to selection bias [30].

Selection bias is a common problem in surveys and is likely to have occurred in this study; we cannot exclude that those who responded to the survey were the most motivated and therefore, those with the best knowledge on antibiotic use. 

Another limitation of this study, intrinsic to its design, is that we were not able to establish associations between the factors possibly related to treatment decisions and the treatment decisions themselves. More research is needed to establish the relationship between knowledge and attitudes on antibiotics as well as resistance and actual prescriptions, so that the gained knowledge could serve to develop evidence-based context-specific training activities as described elsewhere [31,32,33].

Another limitation is the study sample: as it is a convenient, non-random sample, it is not entirely representative of a definite population of physicians, although it is noticeable that the sample of GPs and PDs we reached does not differ in a meaningful way by age and gender from the study population [34,35,36]. At the same time, due to the high number of respondents, we believe that our explorative findings could be useful to guide future studies in the field that should target larger and more representative samples of physicians.

It has to be mentioned that we chose to ask only a few knowledge questions on each explored topic (lower respiratory tract infections, otitis, skin infections, antibiotic duration etc.) rather than focusing in depth on one or a few specific topics and this could have limited the completeness of our findings. 

Finally, items that explored knowledge were dichotomous, therefore, some respondents could have guessed the right answers, so that over- or underestimation of knowledge among the two groups is possible.

## 4. Materials and Methods

### 4.1. Questionnaire Development and Design

As to content validation, questionnaires were constructed in a multidisciplinary team of experts of the Department of General Practice, the Institute for Hygiene and Microbiology and the Department of Pediatrics of the University Hospital Würzburg. To assess face validity, two general practitioners and two outpatient pediatricians reviewed the questionnaires to assess if items were clear and comprehensive.

The questionnaires comprise the following six domains: self-confidence (GP/PD: 5 items), knowledge of guideline recommendations (GP: 19/PD: 20 items), subjective influences and situational factors (GP/PD: 8 items), training (GP: 5/PD: 7 items) and sociodemographic data (GP: 8 / PD: 9 items). Table 6 summarizes the survey domains of both questionnaires. The full surveys are retrievable in the Appendix A.

The guideline-knowledge items were formulated in such a way that each could be assigned to one of the following categories: indication, dose, duration and choice of antibacterial agent. Each category included 4 to 6 of the 19 (GP) or 20 (PD) guideline- knowledge items. Self-confidence in antibiotic use was also explored for each of the above four categories. Knowledge items are mainly based on the guidelines of the German College of General Practitioners and Family Physicians [37], the handbook of the German Society for Pediatric Infectious Diseases [38] and based on a previous survey [39].

### 4.2. Study Design and Sampling of Study Population

A cross-sectional survey about antibiotic knowledge was conducted among 2092 GPs in the Bavarian region of Franconia and among 1661 pediatricians in Bavaria and Baden-Wuerttemberg, which were all the GPs and PDs of the included regions. Data collection took place for 12 weeks, from January to April 2019 for GPs and from May to August 2019 for PDs. The invitation to complete the survey was mailed to all eligible GPs and PDs. A reminder for participation in the survey was mailed at a six-week interval after the first and both were enclosed in an official envelope with the logo of the University department and were accompanied by an invitation letter, a brief instruction and a prepaid addressed envelope for returning the completed questionnaire. An online participation was possible by accessing the survey through a link and a five-digit access code, both printed on the first page of the questionnaire; for PDs, it was additionally possible to access the link via a QR code printed on the cover letter. Each access code could be used only once.

### 4.3. Inclusion Criteria

The eligible study population comprised all GPs listed by the Bavarian Association of Statutory Health Insurance Physicians (KVB) in the region of Franconia and all PDs listed in both, the KVB and the National Association of Statutory Health Insurance of Baden-Württemberg (KVBW) homepages. The federate state Baden-Wuerttemberg was chosen, in addition to Bavaria, as it is comparable according to location, sociodemographic characteristics and antibiotic prescribing rates [6]. The search in the database resulted in 2093 GPs in Franconia (accessed: November 14, 2018) and 746 PDs in Bavaria and 934 in Baden-Wuerttemberg (accessed: March 17, 2019). If a physician was listed in more than one location, only his primary address was considered). In round 1, 2093 GPs and 1680 PDs were contacted via mail. During the first round, 10 more GPs who were not listed requested study documents and were included in round 2. Based on the undelivered letters after round 1, the address lists were revised: The correct addresses of the letters that had not arrived were determined via internet research. In some cases, practice closures turned out to be the reason; these practices were subsequently no longer contacted. Taking these considerations into account, we assume a final reachable sample of 2092 GPs and 1661 pediatricians. The sum is used as the basis for calculating the overall response rate (reachable practices: n = 3753).

The study was advertised among GPs by circular fax by the Bavarian Family Physician Association (BHÄV). Among PDs, a circular fax was sent by the Professional Association of Pediatricians (BVKJ).

### 4.4. Statistical Analysis

The sociodemographic variables queried in categories are presented as proportions. In some cases, categories were combined for greater clarity (e.g., age). Means and standard deviations were calculated where appropriate.

Originally, the questionnaire for GPs also contained 20 knowledge questions. Due to incorrect formulation of one item, only 19 items were considered for analysis. An overall knowledge score was calculated by adding up the correct answers. Nine percent (n = 55/630) of GPs and 9% of PDs (n = 64/681) omitted one question of the 19 and 20 knowledge items, respectively. The missing answer was counted as incorrect and included in the calculation of the total knowledge score.

The relationship between the knowledge score and self-confidence in prescribing antibiotics (item 1.1) in each population (GP or PD) was examined using Spearman’s correlation coefficient.

Two-sample t-tests were performed to assess differences of responses on a 7-point Likert scale between GPs and PDs. In case of violations of variance homogeneity determined with Levene’s test, Welch’s t-test was used instead. Cohen’s d was calculated as a measure of effect size of these differences and interpreted as follows: low if d < 0.5, medium if d ≥ 0.5 < 0.8 and high if d ≥ 0.8.

To assess the relationship in answering knowledge questions according to medical specialty (coding: GP = 1; PD = 0), univariate logistic regressions were performed for each of the seven identical knowledge questions (coding: right answer = 1, wrong answer = 0). As a measure of the effect size, the odds ratio (OR) is given together with the 95% confidence interval (CI). For all statistical tests, the significance level was set to 5% (*p* < 0.05).

Analyses were carried out with IBM SPSS v26.0 [40]. Cohen’s d effect sizes were calculated using JASP [41].

## 5. Conclusions

This large-scaled survey conducted among primary care physicians in southern Germany shows that guideline-knowledge and confidence in antibiotic therapy decisions is very high but awareness about the influence of patients’ expectations on prescribing decisions is low. At the same time, we show that the demand for training in the field of antimicrobial treatment is high. More studies are needed to explore how consultation-related and context-specific factors may influence antibiotic prescriptions in general and pediatric primary care in Germany beyond knowledge. Moreover, more research is needed to explore the training needs of primary care physicians in Germany, as this would serve the development of evidence-based educational interventions targeted to the improvement of antibiotic prescribing decisions rather than being focused solely on guideline-knowledge.

## Figures and Tables

**Figure 1 antibiotics-10-01131-f001:**
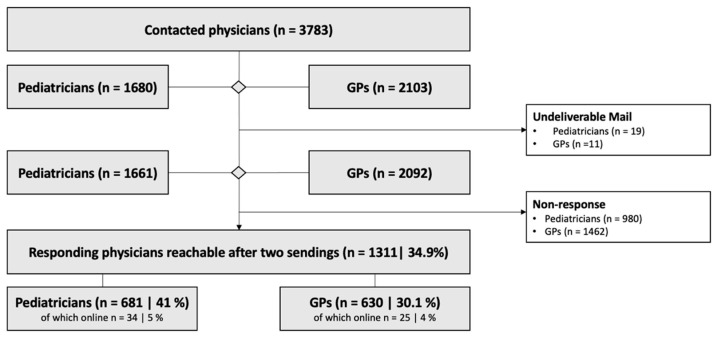
Flow diagram of participants through the study. Ten out of 2103 GPs requested the study documents themselves and were not originally included in the contact list.

**Table 1 antibiotics-10-01131-t001:** Sociodemographic characteristics of participants.

Variables	GP n (%)	PD n (%)
**Gender ^a^**		
Male	332 (62.4)	340 (56.1)
Female	200 (37.6)	266 (43.9)
**Age group ^b^**		
Below 50	144 (24.7)	195 (29.2)
50–59	237 (40.6)	314 (47.1)
60 and above	203 (34.8)	158 (23.7)
**Location ^c^**		
Rural area	221 (37.9)	38 (5.7)
Small town	191 (32.8)	244 (36.7)
City	68 (11.7)	224 (33.7)
Large City	103 (17.7)	159 (23.9)
**Structure ^d^**		
Single practice	269 (46.1)	295 (44.7)
Joint practice	298 (51.0)	347 (52.6)
Medical care centers	17 (2.9)	18 (2.7)
**Number of doctors ^e^**		
1	176 (32.0)	173 (26.1)
2	172 (31.3)	217 (32.7)
3 and above	202 (36.7)	274 (41.3)

^a^ Missing: 98 (GP), 75 (PD). ^b^ Missing: 46 (GP), 14 (PD). ^c^ Missing: 47 (GP), 16 (PD). ^d^ Missing: 46 (GP), 21 (PD). ^e^ Missing: 80 (GP), 17 (PD).

**Table 2 antibiotics-10-01131-t002:** Univariate logistic regression analysis of knowledge differences between GPs and PDs.

	Category	Total		PD		GP			
Topic		n	%	n	%	n	%	OR (95%CI)	*p*
		Correct ^1^		Correct ^1^		Correct ^1^			
Respiratory tract infections	Indication	1302	97	680	98	622	97	0.91 (0.46–1.80)	0.792
Otitis media	Indication	1292	92	673	96	619	87	0.30 (0.19–0.47)	<0.001
Tonsillitis	Agent	1293	91	670	96	623	86	0.28 (0.18–0.43)	<0.001
Pneumonia	Agent	1299	88	678	90	621	86	0.68 (0.48–0.95)	0.023
Erythema migrans	Dose	1276	85	652	79	624	92	3.21 (2.27–4.54)	<0.001
Tonsillitis	Duration	1292	77	669	83	623	69	0.46 (0.35–0.60)	<0.001
Resistance formation	Duration	1298	62	675	59	623	66	1.34 (1.09–1.71)	0.007

Results are expressed in absolute values, percentages and OR (Medical specialty was coded as: 0 = PD, 1 = GP). Abbreviations: OR = Odds Ratio, CI = Confidence Interval. ^1^ Percentage of correct responders and number of total responders to this item.

**Table 3 antibiotics-10-01131-t003:** Comparison of levels of confidence in antibiotic use in GPs and PDs and factors reported as relevant for treatment decision/choice.

Variables	GP		PD		Cohen’s d (95% CI)
	Mean (SD)	n	Mean (SD)	n	
**Confidence**					
Overall **	5.8 (1.0)	622	6.3 (0.9)	681	0.5 (0.4–0.6)
Indication **	5.9 (0.9)	622	6.3 (0.9)	681	0.4 (0.3–0.5)
Agent **	5.8 (1.0)	621	6.2 (0.9)	680	0.4 (0.3–0.5)
Duration **	5.9 (1.0)	623	6.2 (0.9)	680	0.3 (0.2–0.4)
Dose **	6.2 (0.8)	622	6.4 (0.8)	679	0.3 (0.2–0.4)
**Factors influencing decisions to prescribe antibiotics**					
Guideline recommendations **	5.6 (1.2)	625	6.3 (0.9)	677	0.7 (0.5–0.8)
Clinical experience *	5.9 (1.0)	625	6.0 (1.1)	678	0.1 (0.0–0.2)
Recommendation from colleagues *	3.5 (1.7)	621	3.2 (1.7)	674	0.2 (0.1–0.3)
Patients’ wishes **	2.2 (1.3)	626	1.9 (1.2)	678	0.2 (0.1–0.3)
Demanding patients **	3.0 (1.6)	627	2.4 (1.4)	678	0.4 (0.3–0.5)
Expected worsening of symptoms **	3.4 (1.6)	627	3.0 (1.5)	677	0.6 (0.1–0.4)
Full waiting room **	1.6 (1.1)	628	1.4 (1.0)	677	0.2 (0.0–0.3)

** *p* < 0.001, * *p* < 0.05. Results of the analysis are shown as mean scores (SD) of the 7-point Likert-scaled items.

**Table 4 antibiotics-10-01131-t004:** Participation and wish for specific training in antibiotic therapy among GPs and PDs.

Item	Total		PD		GP			
	n	% Yes	n	% Yes	n	% Yes	OR (95% CI)	*p*
Participation in training in past 3 years	1305	81.1%	679	84.8%	626	77.2%	0.60 (0.46–0.80)	<0.001
Wish for more training opportunities	1281	73.8%	662	66.8%	619	81.4%	2.18 (1.68–2.83)	<0.001

Results are expressed in absolute values, percentages and OR (Medical specialty was coded as: 0 = PD, 1 = GP). Abbreviations: OR = Odds Ratio, CI = Confidence Interval.

**Table 5 antibiotics-10-01131-t005:** Perceived need for training offers among GPs and PDs.

Item	PD		GP	Cohen’s d (95% CI)	
	Mean	n	Mean (SD)	n	
Perceived sufficiency of training opportunities **	4.6 (1.5)	672	3.7 (1.6)	613	0.6 (0.5–0.7)

** *p* < 0.001. Results of the analysis are expressed as mean scores (SD) of the 7-point Likert-scaled items among GPs and PDs.

**Table 6 antibiotics-10-01131-t006:** Survey domains, exemplifying items and response format.

Topic	Exemplifying Item	Response Format
Self-confidence	I feel confident in the choice of an antibiotic agent for the treatment of common infectious diseases.	7-point Likert scale (I don’t agree at all–I totally agree)
Guideline knowledge	The likelihood of resistance development increases with the duration of antibiotic treatment.	Dichotomous (Right or wrong (GPs) agree or disagree (PDs))
Subjective influences	When treating infectious diseases, I follow the guideline recommendations of my specialist society.	7-point Likert scale (I don’t agree at all–I totally agree)
Situational factors	When I find the patient to be very demanding during the consultation, I sometimes give in and prescribe an antibiotic without it being necessarily indicated.	7-point Likert scale (Not at all–very strongly)
Training	Have you participated in one or more training courses on the topic of (rational) antibiotic therapy within the last 3 years?	Mixed (e.g., yes/no)
Socio-demographics	Please indicate your age.	Mixed (e.g., age categories)

## Data Availability

The raw data are available from the corresponding author due to reasonable request.

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
