# Peer review of "Management of Common Infections in German Primary Care: A Cross-Sectional Survey of Knowledge and Confidence among General Practitioners and Outpatient Pediatricians"

_antibiotics, 2021, doi:10.3390/antibiotics10091131_

Round 1
Reviewer 1 Report
In the introduction, please describe why this study is significant for this specific region of southern Germany.
Also, important to define the patient population of the GP vs PD! Do GP in Germany treat kids? That would impact their knowledge and comfort level of antibiotics and prescription pattern.
I also think the reorganization of the manuscript should be strongly considered. #1 Introduction
#2 Material and methods
#3 Results section
#4 Discussion (I would ensure that selection Bias and limitations are clearly defined and labeled).
#5 Conclusion
Author Response
REVIEWER 1.
1) In the introduction, please describe why this study is significant for this specific region of southern Germany.
Please consider that we added “in Germany” (line 63) to our justification on why this study is important in the introduction. We undertook this survey among GPs only in Franconia for practicality, we had to restrict the study population because the study was entirely self-funded (it did not receive any external fundings). As we were interested in a comparison among GPs and PDs, we selected a comparable population of PDs based in Bavaria and Baden-Wuerttemberg. We believe that these choices in the design stage are limitations of the study (cf. discussion) nevertheless we feel that even if our results are explorative, they give a contribution in the explanation of the variability of antibiotic prescribing behavior among PCPs in Germany. To fully explain this variability more studies with different study designs are needed (cf discussion and conclusion).
2) Also, important to define the patient population of the GP vs PD! Do GP in Germany treat kids? That would impact their knowledge and comfort level of antibiotics and prescription pattern.
We agree on this and have amended the introduction accordingly (lines 48-51), adding more information about child and adult medical care performed by GPs and PDs.
3) I also think the reorganization of the manuscript should be strongly considered.
#1 Introduction
#2 Material and methods
#3 Results section
#4 Discussion (I would ensure that selection Bias and limitations are clearly defined and labeled).
#5 Conclusion
Please consider that we had to follow editorial policies. Antibiotics asks authors to use the following sequence of sections: Introduction, results, discussion, materials and methods, conclusion. We would also prefer to organize the article in the section sequence you suggest and if the production office allows, we will
change the sequence accordingly.
3.1) (I would ensure that selection Bias and limitations are clearly defined and labeled).
We reorganized the discussion section accordingly and added 2 paragraphs (comparison to existing literature and strengths and limitations). In the limitations´ paragraph, selection bias is clearly mentioned at the beginning of a section of this paragraph.
Reviewer 2 Report
Dear Editors,
Kurotschka and colleagues present a well-written report regarding knowledge and confidence on antibiotics prescription among German GPs and Peds. Although other works on this subject have been published, there are some very interesting and unique aspects that make this paper suitable for publication. Nevertheless, some flaws need to be addressed.
Point 1:
as the Authors already know (ref 5 and 10), physician’s attitude on antibiotic prescription varies considerably within the country. Nevertheless, they choose to conduct the survey only in a specific region. I think this must be stressed in the limits of this study
Point 2:
a very interesting point of this work is that this population of physicians seem to be less influenced by patients wishes, expectations and demands, compared to other works. I think the Authors should stress more this results.
Author Response
Thank you for your constructive review. Please find below our point-by-point answers.
Point 1: as the Authors already know (ref 5 and 10), physician’s attitude on antibiotic prescription varies considerably within the country. Nevertheless, they choose to conduct the survey only in a specific region. I think this must be stressed in the limits of this study.
We added that future studies should target larger and more representative samples (in addition to the fact that we stated that our findings are explorative and that our sample is a result of convenience, non-random sampling) (line 200 onwards).
Point 2: a very interesting point of this work is that this population of physicians seem to be less influenced by patients wishes, expectations and demands, compared to other works. I think the Authors should stress more this results.
We agree that this finding is interesting. We deal extensively with this finding in the discussion section. Following your suggestion to stress more the result, we added (line 164 onwards) a clear recommendation for training and future studies on the topic.
Reviewer 3 Report
I have read this paper with great interest, i value the effort as made.
Abstract
Is the 70 % applicable to children, or a more general (any patient) antibiotic prescription
Questionnaire
A lot of answers were dichotomous, the ‘I don’t know’ option was not included, so that guessing practices should be considered.
The study reports on how GP and pediatricians perceive their practices and approaches, not necessary their ‘true’ practice (response bias, preferences), so this should perhaps be reflected in the title ? this is marginally mentioned in the discussion, but need some additional reflections.
Not sure that resistance prevention is not the only ‘health benefit’ of avoiding unneeded AB prescription.
We need somewhat more information on the healthcare organization related to child medical care in Germany to better understand the setting (quite different within Europe)
Do you have any ideas on the ‘characteristics’ of the full population targeted versus the responding collegues (age, gender, location, cf table 1)
The development of the questionnaire is not sufficiently well described, and there is not yet anything on (content)-validation ?
Author Response
Thank you for the constructive review. Please find below our point-by-point answers.
- Abstract: Is the 70 % applicable to children, or a more general (any patient) antibiotic prescription
“Almost 70%” is applicable to all patients that consult a PCP, as it is the sum of 59% (GPs´ prescriptions) and 9% (PDs´ prescriptions) (lines 46-48 introduction). We changed the phrasing in the abstract slightly (line 14) to clarify the concept.
- A lot of answers were dichotomous, the ‘I don’t know’ option was not included, so that guessing practices should be considered.
We amended the limitation sections by adding over- or underestimation of knowledge among groups as a possibility (lines 209-211), although we believe that no relevant over- or underestimation occurred, as the results of the application of a simple item guessing correction score did not show significant differences.[1]
- The study reports on how GP and pediatricians perceive their practices and approaches, not necessary their ‘true’ practice (response bias, preferences), so this should perhaps be reflected in the title? this is marginally mentioned in the discussion, but need some additional reflections.
We believe that title and text reflect in a proper way these issues:
- we discussed about the possibility that social desirability could have biased the answers,
- we clearly stated that we could not explore actual practices and that for this reason more studies are needed to explore the association among, on the one hand, knowledge and attitudes and, on the other hand, practices.
- We discussed sources of selection bias
- The title, in our view, does not suggest that we measured practices
- Not sure that resistance prevention is not the only ‘health benefit’ of avoiding unneeded AB prescription.
We agree and have amended the introduction accordingly (lines 38-40), mentioning more health benefits of avoiding unnecessary prescriptions.
- We need somewhat more information on the healthcare organization related to child medical care in Germany to better understand the setting (quite different within Europe)
We agree on this and have amended the introduction accordingly (lines 48-51), adding more information about child and adult medical care performed by GPs and PDs.
- Do you have any ideas on the ‘characteristics’ of the full population targeted versus the responding collegues (age, gender, location, cf table 1)
This is a relevant question. We could not retrieve detailed official information about practice locations but we compared the age and sex distribution in our sample with that of the source population of GPs and PDs and amended the discussion accordingly (lines 200-203).
- The development of the questionnaire is not sufficiently well described, and there is not yet anything on (content)-validation?
Details on content- and face- validation were added in the methods section (lines 215-219).
- Frary, R.B. Formula Scoring of Multiple-Choice Tests (Correction for Guessing). Educational Measurement: Issues and Practice 1988, 7, 33-38, doi:https://doi.org/10.1111/j.1745-3992.1988.tb00434.x.